# RNA Helicase DDX3: A Double-Edged Sword for Viral Replication and Immune Signaling

**DOI:** 10.3390/microorganisms9061206

**Published:** 2021-06-03

**Authors:** Tomás Hernández-Díaz, Fernando Valiente-Echeverría, Ricardo Soto-Rifo

**Affiliations:** 1Laboratory of Molecular and Cellular Virology, Virology Program, Institute of Biomedical Sciences, Faculty of Medicine, Universidad de Chile, Santiago 8380453, Chile; tomas.hernandez@ug.uchile.cl (T.H.-D.); fvaliente@uchile.cl (F.V.-E.); 2HIV/AIDS Workgroup, Faculty of Medicine, Universidad de Chile, Santiago 8380453, Chile

**Keywords:** DDX3, mRNA metabolism, type-I interferon, viral replication, antiviral target

## Abstract

DDX3 is a cellular ATP-dependent RNA helicase involved in different aspects of RNA metabolism ranging from transcription to translation and therefore, DDX3 participates in the regulation of key cellular processes including cell cycle progression, apoptosis, cancer and the antiviral immune response leading to type-I interferon production. DDX3 has also been described as an essential cellular factor for the replication of different viruses, including important human threats such HIV-1 or HCV, and different small molecules targeting DDX3 activity have been developed. Indeed, increasing evidence suggests that DDX3 can be considered not only a promising but also a viable target for anticancer and antiviral treatments. In this review, we summarize distinct functional aspects of DDX3 focusing on its participation as a double-edged sword in the host immune response and in the replication cycle of different viruses.

## 1. Introduction

DEAD-box polypeptide 3, X-linked or DDX3X (hereafter referred as DDX3) belongs to the DEAD (Asp-Glu-Ala-Asp) box family of ATP-dependent RNA helicases present in various eukaryotic organisms from yeast to humans [1,2]. This enzyme is a multifunctional RNA-binding protein playing roles in various aspects of the mRNA metabolism, including transcription, splicing, nuclear export, translation and localization [3,4,5,6,7]. However, DDX3 has also been implicated in various cellular processes, such as the regulation of cell cycle, cancer progression, innate immune response and viral infections [8,9,10].

As an RNA helicase from the DEAD-box family, DDX3 is organized by a *N*-terminal domain, two RecA-like domains that constitute the helicase core and a *C*-terminal domain (Figure 1a). The RecA-like domains contain 12 conserved motifs involved in ATP binding, RNA binding and linking ATP hydrolysis with RNA unwinding where the enzymatic reactions occur (Figure 1b) [11,12,13,14]. The helicase core is well conserved in multiple helicases and mostly in the yeast orthologue Ded1, which has been useful as a model for studying the structure and function of DDX3 [15,16]. The N- and the *C*-terminal domains are variable amongst DEAD-box helicases, and it has been proposed that both external domains of DDX3 regulate protein-protein interactions or confer RNA substrate specificity to DDX3 (Figure 1b) [2,12,13,17]. 

DDX3 can form dimers and interact with RNA and proteins, forming different complexes to modulate their function and, interestingly, while some functions of DDX3 are dependent on its catalytic activity, others only depend on its direct interaction with RNA or protein partners [18]. Particularly, during viral infections, DDX3 has been observed to play a dual role in viral replication: as a viral RNA sensor and mediator of the innate immune response but also as a cellular factor promoting viral replication [19,20].

## 2. Role of DDX3 on mRNA Metabolism

DDX3 has been described as presenting nuclear and cytoplasmic localization where it fulfills different functions associated with mRNA metabolism [21]. Inside the nucleus, DDX3 can act as a co-transcriptional factor to regulate transcription of a subset of genes, essentially associated with cell cycle progression and innate immune response [4,9,22]. Also, DDX3 associates with messenger ribonucleoproteins (mRNPs), predominantly with spliced mRNAs carrying an exon junction complex (EJC) [5,6]. Association of DDX3 with nuclear mRNPs occurs through an interaction with some of the core EJC proteins and the nuclear cap-binding complex (CBC). This association requires both splicing and the deposition of the EJC upstream of an exon–exon junction but not the binding of the CBC to the cap structure [6,23].

Also, DDX3 is a nucleo-cytoplasmic shuttling protein that has been associated with different mRNA nuclear export pathways. As such, DDX3 was shown to interact with nuclear export factor 1 (NXF1/TAP) and is exported along with spliced messenger ribonucleoprotein complexes (mRNPs) to promote specific mRNA translation [5,24,25]. Another nuclear export pathway in which DDX3 has been involved is that driven by chromosome region maintenance 1 (CRM1), a RanGTP-dependent system that participates in the export of nuclear export signal (NES)-containing cargoes as well as snRNA and some specific mRNA [26,27]. DDX3 interacts with CRM1 through a NES relocating DDX3 into the cytoplasm (Figure 2a) [23,28]. Of note, as will be discussed below, the interaction between DDX3 and CRM1 is exploited by HIV-1 to favor the cytoplasmic accumulation of intron-containing viral RNA [29,30,31].

Another aspect of mRNA metabolism in which DDX3 has been involved is translation, specifically at the initiation step. As such, many studies have shown that DDX3 depletion did not affect global protein synthesis but affect a small subset of the transcriptome. In particular, this subset of mRNA sensitive to DDX3 depletion has a highly structured 5′-untranslated region (5′-UTR) and possesses an RNA-protein specific interaction with DDX3 [5,10,32]. Therefore, it seems that DDX3 does not participate in global protein synthesis but rather it promotes the efficient translation of a subset of specific mRNAs carrying structured 5′-UTR through its helicase activity, which can facilitate translation through the resolution of secondary structures during ribosomal scanning (Figure 2b) [33]. Moreover, DDX3 may promote the entry of specific mRNAs in translation initiation through local remodeling of secondary structures near the cap prior 43S ribosomal scanning (Figure 2c) [32,33,34,35].

Indeed, DDX3 may actively regulate translation initiation through interactions with multiple translation initiation factors. As such, DDX3 could act as an inhibitor of cap-dependent translation by disrupting the eIF4E/eIF4G interaction through a YXXXXLϕ eIF4E-binding motif present at the *N*-terminal domain [36]. Furthermore, cap affinity chromatography analysis suggests that DDX3 traps eIF4E in a translationally inactive complex by blocking its interaction with eIF4G [7,36,37]. Moreover, it was shown that point mutations within the consensus eIF4E-binding motif present in DDX3 impair its binding to eIF4E, resulting in a loss of DDX3′s regulatory effects on translation initiation (Figure 2d) [36]. Similarly, DDX3 enhanced the association of the cap-binding complex (CBC), with upstream open reading frames (uORF) containing mRNAs facilitating the recruitment of eIF3 [38]. DDX3 also interacts with eIF3B and eIF3C subunits and this interaction has been linked to the expression of metastatic genes [38,39,40]. Additionally, it has been described that eIF4F, eIF4A and EIF2S1 are other initiation translation factors that interact with DDX3 in an RNA-independent manner to promote translation of selected mRNAs [5,32,39].

In an alternative way to promote translation initiation, DDX3 can facilitate the assembly of 80S ribosomes and then be released from ribosomes in an ATPase activity independent manner before translation elongation begins (Figure 2e) [39]. All this evidence shows that DDX3 participates exclusively in translation initiation through distinct interaction with the translation initiation machinery including mRNAs, translation initiation factors and ribosomal subunits to remodel gene expression programs in response to different cellular conditions [41,42].

An additional way shown to regulate protein synthesis and in which DDX3 has been involved is through the assembly of stress granules, large ribonucleoprotein complexes containing mRNA stalled in translation initiation together with several RNA-binding proteins [43,44,45,46]. These membraneless organelles are highly dynamic and form upon a variety of stressful conditions that result in the inhibition of translation initiation functioning as sites where mRNA triage and storage takes place [44,45,47]. DDX3 has been found as a component of cytoplasmic stress granules, its overexpression triggers the spontaneous assembly of these structures, and its pharmacological inhibition attenuates stress granules formation, suggesting that DDX3 is an important regulator of SG assembly and maturation, concordant with its ability to interact with many components of the translation initiation machinery [7,46,48,49]. In particular, DDX3-mediated SG assembly may affect translation of mRNAs bearing upstream open reading frames and probably DDX3-mediated cap-independent translation [38,50].

## 3. Role of DDX3 in Immune Signaling Pathways

Recognition of foreign nucleic acids by sensor proteins able to distinguish between exogenous and self-genetic material is essential for a robust host defense against invading pathogens [51]. Sensing of viral nucleic acids during infection triggers different antiviral responses, including the activation of NLRP3 inflammasome, the type-I interferon (IFN-I) response and the programmed cell death activation, in order to restrict viral replication [52]. It is noteworthy that the RNA helicase DDX3 has been shown to participate in all these pathogen sensing pathways.

First, DDX3 regulates NLRP3 inflammasome activation leading to the induction of IL-1β and pyroptosis [53]. Interestingly, it was shown that the NLRP3 inflammasome and stress granules compete for DDX3 to set the innate immune response and cell death or to promote cell survival against a stress stimulus. Therefore, DDX3 acts as a checkpoint between survival and cell death (Figure 3a) [53,54,55]. Moreover, silencing of DDX3 decreases the translational efficiency of target mRNAs linked to the inflammatory response such as PACT, STAT, Rac1 and TAK1 [34,56].

Similar to RIG-I like receptors (RLR) such as RIG-I, MDA-5 or LGP2, DDX3 acts as a viral RNA sensor that triggers the MAVS-dependent signaling leading to type-I IFN production [19,57,58]. Indeed, DDX3 participates in various steps of the signaling cascade, acting independently of RIG-I and MDA-5 through a direct association with MAVS [59,60,61]. It has been shown that after associating with MAVS, DDX3 binds IKKε/TBK-1 triggering activation of the kinase activity of the complex. Here, IKKε became autophosphorylated and then phosphorylates serine residues present at the *N*-terminal domain of DDX3. Phosphorylated DDX3 recruits IRF3/IRF7, which are also phosphorylated by IKKε/TBK-1 [62]. In this pathway, DDX3 acts as a bridge that allows IKKε to phosphorylate IRF3, leading to IFN production [22,63,64]. Alternatively, phosphorylation of DDX3 by TBK-1 allows the translocation of the RNA helicase to the nucleus where it binds to the type I IFN promoter, leading to transcriptional activation of IFN-α/β genes (Figure 3b) [22,65]. A common characteristic of these different type-I IFN-stimulating pathways is that the enzymatic activity of DDX3 is not required [22,66].

## 4. DDX3 as a Host Factor Involved in Viral Replication

Despite all its functions in immune activation described above, DDX3 has been described for several years as a cellular factor essential for the replication of different viruses (Table 1). These viruses include several RNA viruses including murine norovirus (MNV), some members of the *Flaviviridae* family such as West Nile virus (WNV), dengue virus (DENV), Japanese encephalitis virus (JEV) and hepatitis C virus (HCV), picornaviruses (EV71 and FMDV), alphavirus Venezuelan Equine Encephalitis Virus (VEEV), human immunodeficiency virus type 1 (HIV-1), arenaviruses (JCMV and LUNV), influenza virus A (IAV) and human parainfluenza virus-3 (HPIV-3), but also DNA viruses such as herpes simplex virus type 1 (HSV-1), human and murine cytomegalovirus (HCMV and MCMV) and hepatitis B virus [20]. Given the multifaceted functions of DDX3 linked to mRNA metabolism and the innate immune response, several viruses use DDX3 to promote their replication cycle either by promoting viral gene expression, by inhibiting the antiviral cellular response or both.

### 4.1. Norovirus

Norovirus belongs to the *Caliciviridae* family and possesses a positive sense single stranded RNA molecule as a genome. Given the lack of a cell culture model for human norovirus, most of the studies of norovirus replication have been carried out with murine norovirus (MNV) [67,68]. DDX3 was reported as a viral RNA-binding protein that accumulates in MNV cytoplasmic replication sites and its knockdown results in a strong reduction of viral RNA and protein synthesis as well as viral titers [69].

### 4.2. Flaviviruses and Hepacivirus

Members of the *Flaviviridae* family are enveloped viruses with a positive-sense single stranded RNA molecule as a genome [70]. West Nile virus (WNV) is a flavivirus that hijacks DDX3 from P-bodies to cytoplasmic sites of viral replication, altering the assembly of these membraneless organelles while promoting viral replication [71]. In the case of Japanese encephalitis virus (JEV), DDX3 was shown to favor viral replication through the binding to the 5′-UTR of the viral genome and to viral proteins NS3 and NS5 [72]. More recently, DDX3 was shown to unwind the 5´-UTR of the viral genome suggesting that the cellular RNA helicase facilitates viral RNA translation and consequently viral replication [73]. In the same way, DDX3 was shown to unwind the 5´-UTR of the Zika virus (ZIKV) genome [73].

Evidence for a role of DDX3 during dengue virus (DENV) replication is more divergent. It has been described that DDX3 inhibits viral replication through the induction of the type-I IFN antiviral signaling pathway [74]. In agreement with this observation, other experiments showed that DENV Capsid protein interacts with DDX3 to abolish the antiviral functions of the RNA helicase [75]. In addition, during the late stages of infection, DENV down-regulates DDX3 expression to facilitate viral replication [75]. However, treatment with the DDX3 inhibitor RK-33 reduces DENV replication, suggesting that DDX3 promotes viral replication [76,77]. Furthermore, some evidence showed that DDX3 interacts with the calcium channel TRPV4 to facilitate viral replication of DENV and ZIKV [78]. In both cases, it is necessary to carry out more studies to determine whether or not DDX3 promotes DENV and ZIKV replication.

More detailed evidence exists for the involvement of DDX3 during the replication of the hepacivirus, hepatitis C virus (HCV), a virus with an important relevance in global public health. First, it was demonstrated that DDX3 interacts with the *C*-terminal domain of HCV core protein and the knockdown of DDX3 reduces HCV RNA and core protein levels [79,80]. Interestingly, the DDX3-core interaction seems to promote HCV replication in a genotype-dependent way, with HCV genotype 1 being favored while other genotypes are not affected [81]. Moreover, HCV core inhibits the interaction between DDX3 and MAVS while the viral NS3 protease induces degradation of MAVS with the consequent reduction of IFN-β production and the stimulation of viral replication [82,83].

The HCV core protein also relocates DDX3 to lipid droplets (LD), lipid-rich structures derived from endoplasmic reticulum in which several viral proteins associate to drive RNA synthesis and viral particle assembly. Interestingly, recognition of the viral RNA by DDX3 triggers LD biogenesis in a pathway involving the relocalization of IKKα and SREBP activation [84,85]. Finally, it has been observed that HCV infection promotes stress granules assembly and DDX3 binding to the 5′-UTR of the HCV RNA promotes recruitment of viral RNA to stress granules and LD to carry out viral replication. As such, inhibition of DDX3 reduces stress granules assembly limiting viral replication (Figure 4) [81,85,86].

### 4.3. Picornaviruses

Enterovirus 71 (EV71) and foot and mouth disease virus (FMDV) belongs to the *Picornaviridae* family and causes a self-limiting infection with occasionally neurological complications such as encephalitis, mainly in children [87,88]. Both viruses possess a positive sense single stranded RNA molecule as a genome and contains an internal ribosome entry site (IRES) within their 5′-UTR [87,89]. For EV71, it has been described that DDX3 positively regulates translation in an IRES-dependent manner. Here, EV71 IRES contain a secondary structure that requires the helicase activity of DDX3 to promotes EV71 RNA translation and consequently, when the secondary structure of EV71 IRES is destabilized, or DDX3 is knocked down or lacks helicase activity, viral replication decreases [89]. In the case of FMDV, to carry on its replication, it has been shown that it requires ribosome protein RPL13 binding to the viral IRES, and this binding between RPL13 and FMDV IRES is dependent on DDX3. Also, DDX3 and RPL13 cooperates to support the assembly of 80S ribosomes and promote translation initiation of viral mRNA. Here, DDX3 participates in viral replication independent of its enzymatic activities, thus relying on a protein-protein interaction with RPL13 [90].

### 4.4. Alphavirus

Venezuelan equine encephalitis virus (VEEV) is a mosquito-borne New World alphavirus with no current approved vaccines or treatments [91,92]. VEEV is an enveloped virus with a positive sense single stranded RNA genome that encodes structural and non-structural proteins. Particularly, non-structural protein 3 (nsP3) presents hypervariable domains that mediate interactions with host proteins [93,94]. Within the VEEV replication cycle, it has been described that the nsP3 protein interacts with RNA helicases DDX1 and DDX3 to promote viral replication. The mechanism involves a proviral role of DDX3 in the VEEV life cycle in which nsP3 interacts with the translation initiation machinery that includes DDX3, eIF4A, eIF4G and PABP to favor viral protein synthesis [94,95].

### 4.5. Human Immunodeficiency Virus Type-1

HIV-1 is a prototype member of the *Retroviridae* family and the etiologic agent of acquired immunodeficiency syndrome (AIDS), a chronic infectious disease affecting 38 million people worldwide [96,97]. The virus possesses a positive sense single stranded RNA as a genome that is converted into a double stranded DNA and integrated into a host chromosome during infection [98,99]. HIV-1 gene expression is a complex and tightly regulated process, leading to the synthesis of fifteen proteins from a single transcribed RNA, the full-length RNA [100,101,102]. DDX3 takes special relevance for HIV-1 gene expression by promoting nuclear export and translation initiation of the intron-containing full-length RNA both processes requiring the enzymatic activity of the RNA helicase (Figure 5a) [29,30]. As mentioned above, DDX3 contains a NES within its *N*-terminal domain that allows its recruitment into the complex formed by the viral protein Rev and the cellular karyopherin CRM1, which drives the nuclear export of the full-length RNA as well as other intron-containing viral transcripts [29,30]. However, the participation of DDX3 during HIV-1 replication is not limited to nuclear export of a subset of viral transcripts, since the host RNA helicase also participates in translation initiation of the full-length RNA by facilitating the unwinding of RNA structures present close to the m^7^GTP cap structure and the loading of the 43S preinitiation complex in an ATPase and helicase activity-dependent manner [29,32,103]. Finally, it has been shown that DDX3X forms a complex with the viral protein Tat to modulate gene expression of HIV-1 and facilitates Tat-dependent translation of viral mRNAs [104,105]. Consistently with a critical role of DDX3 during the HIV-1 life cycle, knockdown or inhibition of DDX3 has been described to restrict HIV-1 replication without producing cellular death [103,106,107].

Together with its roles as a proviral factor, DDX3 has also been shown to sense abortive HIV-1 RNA transcripts and trigger a MAVS-dependent type-I interferon signaling in dendritic cells (DCs) [59,108,109]. Consistent with its ability to recognize the 5´end of the HIV-1 full-length RNA [32], DDX3 has been shown to sense a synthetic RNA carrying a m^7^GTP cap structure followed by the first 58 nucleotides of the HIV-1 full-length RNA (the TAR RNA structure) and activate IRF3 and NF-κB, leading to DCs maturation and the expression of pro-inflammatory cytokines (Figure 5b) [109]. However, during viral replication, this DDX3/MAVS-driven immune signaling was blocked through the DC-SIGN/PLK1 pathway, which resulted in an accelerated viral replication [59].

More recent work placed DDX3 as a pharmacological target to reduce the HIV-1 latent reservoir. By targeting DDX3 in different models of latency including primary CD4^+^ T cells from people living with HIV under suppressive antiretroviral therapy, it was shown that DDX3 inhibition leads to viral RNA synthesis and IFN-I and NFκB activation, which in turns resulted in the specific apoptosis of HIV-1 carrying cells (Figure 5c) [110]. These new data strongly support the notion that DDX3 is a promising pharmacological target to treat HIV-1 infection at multiple levels.

### 4.6. Paramyxovirus

Human parainfluenza virus type 3 (HPIV-3) belongs to the *Paramyxoviridae* family, and its genome consists of a negative sense single stranded RNA molecule [111]. It has been shown that HPIV-3 infection induces the rapid relocalization of DDX3 from the cytoplasm to the nucleus, resulting in increased IFN-β secretion, and surprisingly, elevated HPIV-3 virus production [28]. Additionally, other investigations showed that the enzymatic inhibition of DDX3 reduces HPIV-3 RNA and viral titers, supporting the proviral role of DDX3 during HPIV-3 replication [77]. However, the mechanism by which DDX3 favors the replication of HPIV-3 is not clear yet and further studies are necessary to elucidate it.

### 4.7. Influenza Virus A

Influenza virus A is a negative sense single stranded segmented RNA virus shown to be inhibited by DDX3. Indeed, it was described that DDX3 interacts with NS1 and NP protein and relocalizes to the viral replicase complex by binding with subunits PB1-F2. This latter interaction has as a consequence, the degradation of DDX3 and viral proteins as a consequence, thus decreasing viral production. Moreover, it has been shown that in the absence of NS1, DDX3 inhibits viral replication through the assembly of stress granules [112,113,114].

### 4.8. Arenaviruses

Arenaviruses are enveloped viruses with a segmented negative sense single stranded RNA as a genome and cause hemorrhagic fever in humans with a mortality rate close to 30% [115]. In the case of arenaviruses JUNV and LCMV, it has been reported that viral NP protein interacts with DDX3, and this interaction has a positive effect on viral RNA synthesis and favors replication through the inhibition of type-I IFN production during infection (Figure 6). These effects were shown to be dependent on the ATPase and helicase activities of DDX3, and as a consequence, DDX3 deficiency has been shown to reduce JUNV and LCMV replication [116].

### 4.9. Vaccinia Virus

VACV, a typical member of the *Poxviridae* family, is a double stranded DNA virus that replicates within cytoplasm [117]. This virus expresses several proteins that facilitate viral replication and avoid the host immune response. The VACV K7 protein is a key virulence factor that binds to the *N*-terminal domain of DDX3 and prevents the activation and induction of the IFN-β promoter, decreasing type-I IFN production (Figure 6) [64,118,119].

### 4.10. Herpesviruses

During herpes simplex virus type-1 (HSV-1) replication, DDX3 was shown to regulate the expression of viral genes and virion assembly [120]. DDX3 was also shown to promote replication of human cytomegalovirus (HCMV) [121]. However, the production of type-I IFN was not altered by infection, since the production of IFN-β is enhanced in HSV-1 and HCMV infected cells [66,121,122]. In contrast, murine cytomegalovirus (MCMV) encodes m139, a protein that targets DDX3 to block type-I interferon production and promote viral replication (Figure 6) [123].

### 4.11. Hepatitis B Virus

HBV is a double stranded DNA virus that also replicates using reverse transcription of viral transcripts. During HBV replication, DDX3 inhibits reverse transcription of the HBV genome by competing with the viral DNA polymerase. However, there is also evidence showing that the HBV DNA polymerase competes with DDX3 for binding to the TBK1/IKKε complex thus, blocking interferon production during infection (Figure 6) [124,125,126].

## 5. DDX3 as a Promising Broad-Spectrum Antiviral Target

Given its multifaceted functions during replication of several viruses, DDX3 emerges as a promising target for the development of broad-spectrum antivirals [77,127,128,129]. Indeed, many different small molecules capable of inhibiting DDX3 enzymatic activities have been developed and evaluated as antiviral molecules [77,130,131,132]. For example, DDX3 enzymatic inhibitors have been evaluated against WNV and DENV infection in vitro, demonstrating antiviral activity with low cellular toxicity and opening the way for new strategies for treating flaviviral infections [76,77,133]. The same occurs with HIV-1 infections where the inability to eradicate the virus and the possibility of viral resistance to antiviral treatment are current challenges. Due to the critical relevance of DDX3 in HIV-1 gene expression and latency reversal, targeting DDX3 emerges as an attractive alternative strategy to fight against HIV-1 chronic infections. Several compounds that inhibit DDX3 enzymatic activity have been tested during HIV-1 replication in vitro, demonstrating low cellular toxicity and an antiviral effect at low concentrations, even against viral isolates resistant to some classical antiretroviral drugs [134,135,136]. Two DDX3 enzymatic inhibitors, RK-33 and compound b, have shown optimal tolerability and biodistribution at therapeutic levels in several mouse tissues [134,137].

## 6. Concluding Remarks

DDX3 appears as a double-edged sword by acting as an important cellular protein involved in mRNA metabolism and immune signaling but also as a host factor that favors replication of several viruses. The multifaceted functions of DDX3 are location and context dependent, as the RNA helicase can localize in the nucleus to regulate transcription, splicing and nuclear export or in the cytoplasm to support translation initiation and to participate in distinct stress response pathways such as NLPR3 inflammasome activation, stress granules assembly and the innate immune response. Due to its variety of functions, many viruses exploit or inhibit DDX3 to promote their replication cycles. In this way, DDX3 emerges as key host factor useful to study virus-host interactions and to know how, when and where viruses use cellular proteins for their own benefit.

During the stages of viral replication, viral PAMPs and in particular viral RNA must avoid recognition by sensors in order to evade the host cell’s immune response. They do this by interacting with different cellular factors to modulate or usurp their functions and/or to inhibit the cellular antiviral response. In this way, the study of the virus per se is not sufficient, as the way it interacts with the cell it infects also needs to be examined. DDX3, which is a multifunctional cellular protein capable of widely promoting the replication of different viruses, is relevant in this context. The increased requirement for DDX3 observed in many different models of viral infections but also in several cancer cells place this cellular RNA helicase as a key target for pharmaceutical intervention.

Last but not least, the study of the interaction between viruses and the host is important when it becomes necessary to fight against viral infections that represent serious threats to the human population, such as the current COVID-19 pandemic. In this context, DDX3 has been identified as a host factor for SARS-CoV-2 replication that binds the viral RNA genome thus, reinforcing the notion that this RNA helicase represents an attractive target for the development of broad-spectrum antivirals [138,139,140].

## Figures and Tables

**Figure 1 microorganisms-09-01206-f001:**
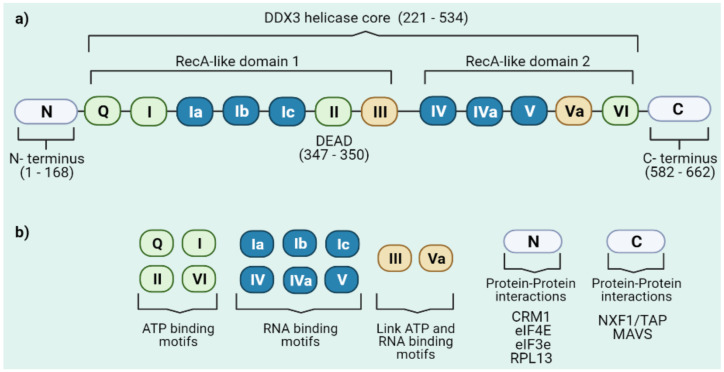
DEAD-box RNA helicase DDX3. (**a**) Schematic representation of the DDX3 organization showing the variable N- and C- terminal domains and the conserved helicase core. The catalytic core consists of two RecA like domains which contain functional motifs. (**b**) Distinct functions of DDX3 motifs. While motifs Q, I, II and VI are involved in ATP binding, motifs Ia, Ib, Ic, IV, IVa and V are involved in RNA binding. Also, motifs III and Va are related to the linking of ATP hydrolysis and double strand unwinding. Finally, DDX3 *N*- and *C*-terminal domains participate in some protein-protein interactions with proteins involved in mRNA metabolism or innate immune signaling.

**Figure 2 microorganisms-09-01206-f002:**
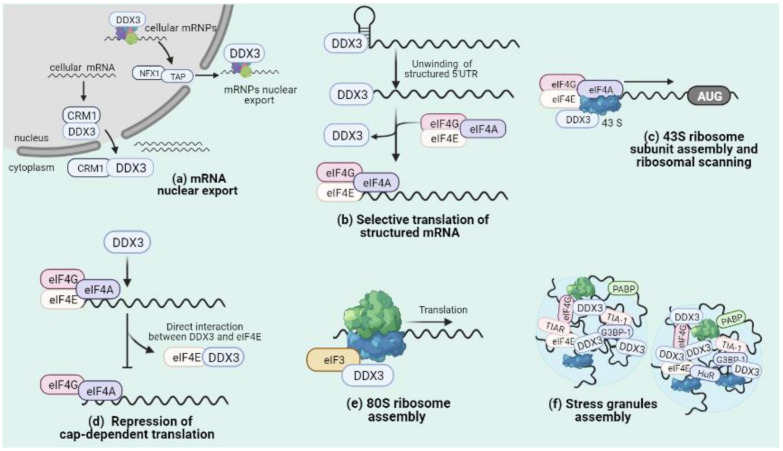
Roles of DDX3 in mRNA metabolism. (**a**) DDX3 participates in NXF1- and CRM1-driven nuclear export pathways to promote the cytoplasmic accumulation of specific subsets of mRNAs and mRNPs. (**b**) DDX3 promotes translation of mRNAs with structured 5′UTR in a helicase activity dependent-manner, through resolution of mRNA secondary structures near of 5′UTR. (**c**) DDX3 participates in 43S ribosome assembly and ribosomal scanning during translation initiation. (**d**) DDX3 interferes with cap-dependent translation through protein-protein interaction with eIF4E and promotes cap-independent translation. (**e**) DDX3 supports 80S ribosome assembly during translation initiation. (**f**) DDX3 is a component of stress granules that participates in their assembly and maturation through RNA-protein and protein-protein interactions.

**Figure 3 microorganisms-09-01206-f003:**
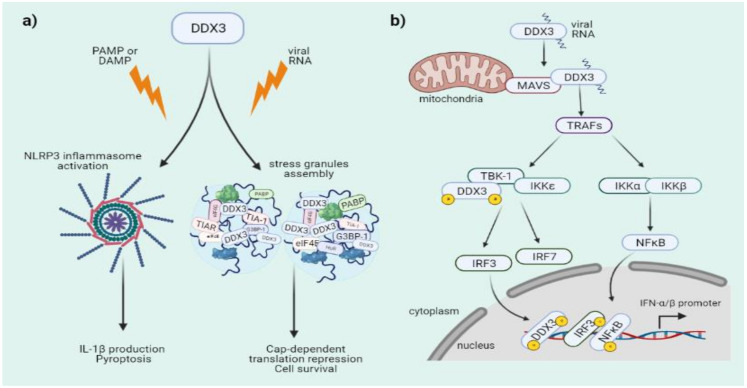
DDX3 participates in immune signaling pathways. (**a**) DDX3 participates in cell fate decisions through NLRP3-inflammasome activation or stress granules assembly. (**b**) DDX3 is a key component during innate immune response participating in sensing of viral RNA and MAVS activation but also as a phosphorylation target and a co-transcriptional factor in the type I IFN production signaling pathway.

**Figure 4 microorganisms-09-01206-f004:**
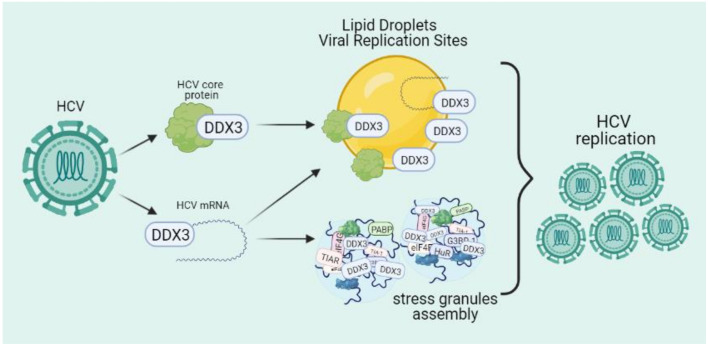
Participation of DDX3 during HCV replication. Two phenomena are described, HCV core protein directly interacts with DDX3 and DDX3 binds the HCV RNA. The interaction between HCV core and DDX3 results in the translocation of DDX3 to lipid droplets and viral replication sites, avoiding type-I IFN pathway activation. On the other hand, DDX3 binds the HCV RNA, inducing stress granules assembly and promoting DDX3 relocalization to lipid droplets to carry out viral assembly and viral replication.

**Figure 5 microorganisms-09-01206-f005:**
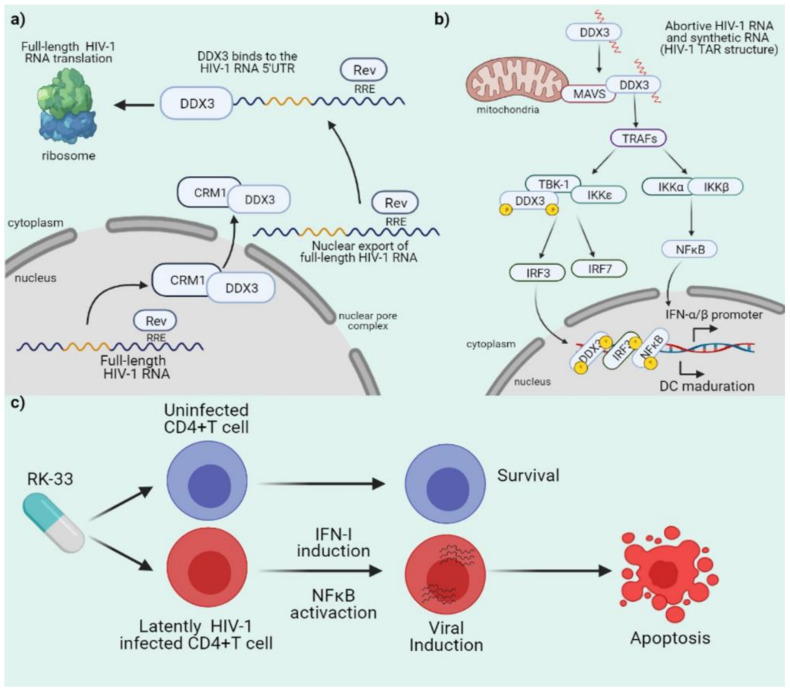
Role of DDX3 during HIV-1 replication. (**a**) DDX3 participates in nuclear export of full-length viral RNA through an interaction with CRM1. Also, DDX3 promotes efficient translation of intron-containing viral RNA through an interaction with the viral RNA 5′ end to promote the synthesis of viral proteins. (**b**) DDX3 binds to abortive HIV-1 RNA and synthetic RNA to promote innate immune response and dendritic cell maturation. (**c**) RK-33 a specific DDX3 inhibitor promotes selective cell death of latently-infected CD4^+^ T cells.

**Figure 6 microorganisms-09-01206-f006:**
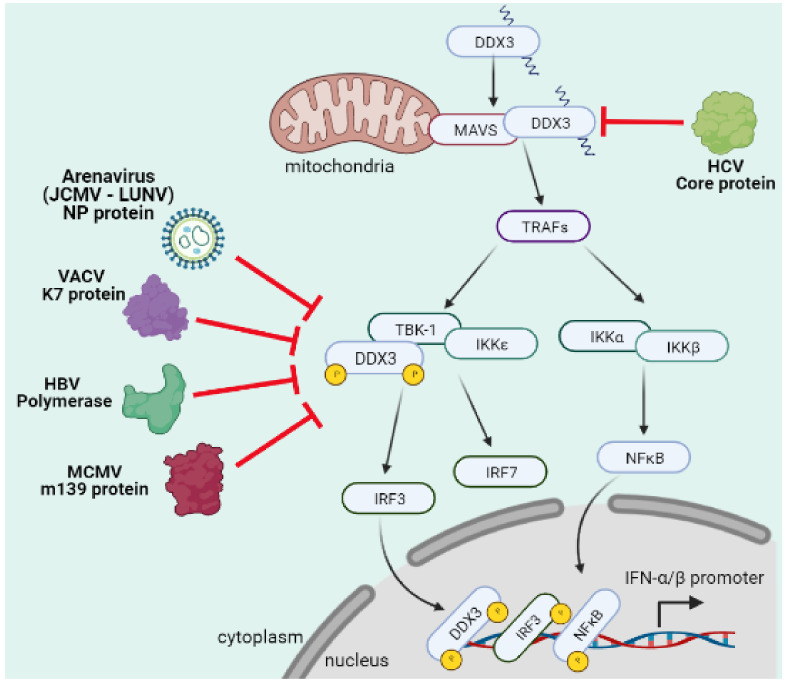
Distinct viral proteins block DDX3 and inhibits type-I IFN production. Arenavirus, VACV K7 protein, HBV polymerase and MCMV m139 protein are able to block interaction of DDX3 with TBK-1/IKKε, avoiding DDX3 phosphorylation and inhibiting type I IFN production. Additionally, HCV core protein interacts with DDX3 and promotes MAVS degradation.

**Table 1 microorganisms-09-01206-t001:** Summary of the interactions between DDX3 and viruses.

Virus	Genome	Viral Interactor of DDX3	Process Regulated by DDX3	Function
MNV	(+) ssRNA	Vral RNA	Viral RNA translation	proviral
WNV	(+) ssRNA	Viral RNA/NS3 protein	Viral RNA translation/block SG assembly	proviral
JEV	(+) ssRNA	Viral RNA/NS3 and NS5 proteins	Viral RNA translation	proviral
ZIKV	(+) ssRNA	Viral RNA	Viral RNA translation	proviral
DENV	(+) ssRNA	Capsid protein	Viral replication/promote IFN-I production	proviral antiviral
HCV	(+) ssRNA	Viral RNA/Core protein	Viral RNA translation/block IFN-I production	proviral
EV71-FMDV	(+) ssRNA	Viral mRNA	Viral RNA translation	proviral
VEEV	(+) ssRNA	nsP3 protein	Viral RNA translation	proviral
HIV-1	(+) ssRNA (RT)	Viral RNA/Tat	Viral RNA export/viral RNA translation	proviral
HPIV-3	(−) ssRNA	Unknown	Viral replication	proviral
IAV	(+) ssRNA	NS1 and NP proteins	Degradation of viral proteins	antiviral
LCMV-JUNV	(−) ssRNA	NP protein	Block IFN-I production	proviral
VACV	dsDNA	K7 protein	Block IFN-I production	proviral
HSV-1-MCMV	dsDNA	m139 protein	Viral expression/Block IFN-I production	proviral
HBV	dsDNA (RT)	DNA polymerase	Block IFN-I production	proviral

## Data Availability

Not applicable.

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
