# Peer review of "RNA Helicase DDX3: A Double-Edged Sword for Viral Replication and Immune Signaling"

_microorganisms, 2021, doi:10.3390/microorganisms9061206_

Round 1

Reviewer 1 Report

The manuscript by Hernandez-Diaz et al. is an interesting and timely review of DDX3/ human virus interplay. The authors describe DDX3 physiological functions in RNA metabolism and innate immune signaling. Then they review how these diverse functions are hijacked and manipulated by RNA and DNA viruses. They describe the interactions engaged between DDX3 and viral proteins and whether they facilitate or instead control virus multiplication. These aspects are clearly synthesized and include recent data from the literature. The review is also very well illustrated, with clear figures that capture the mains aspects described in the review. In this regard, the review is of interest to people interested in the role of this host helicase in viral infections. 

I only have minor suggestions, some concerning the Figures, others about the text.

Main remarks 

The text is very synthetic and some information is lacking. To be fully exhaustive, the list of viruses interacting with DDX3 during their replication cycle should be completed with data dealing with the role of DDX3 in enteroviruses (FMDV), paramyxoviruses (HPIV-3), and alphaviruses (VEEV) life cycle. 

For each viral family, whether the reported role of DDX3 relies on its helicase activity or not should be indicated when possible. 

In the last paragraph, dedicated to DDX3 targeting for therapeutic interventions, it is not clear how anti-DDX3 molecules discussed in this section are acting. They are referred to as “DDX3 inhibitors”. Whether these molecules inhibit DDX3 helicase activity or are inhibiting other DDX3-associated functions needs to be mentioned as well as they potential interest to fight viruses discussed in the previous section. 

Given DDX3's pivotal physiological role in cell RNA fate, an important point relates to the possibility to inhibit DDX3 for antiviral purposes without general adverse effect. The final discussion needs to consider this point. 

Minor remarks on the Figures:

Since the review is focused on DDX3, Figure 1 should depict DDX3 organization instead of that of DEAD-box helicases in general. This especially refers to the exact organization of the N- and the C-terminus domains of DDX3 that should be represented instead of the variable domains indicated in the current figure. The size of each domain should be indicated, as well as binding domains for interacting proteins. The corresponding text needs to be modified accordingly “The N- and the C-terminus domains are variable amongst DEAD-box helicases ...”. 

For more clarity, a Table could be inserted to summarize DDX3 pro- and anti-viral activities in the various models discussed in the review. The nature of viral proteins or nucleic acids interacting with this helicase could be inserted. 

Author Response

Dear Reviewer,

On behalf of the authors and myself, I would like to thank for your comments that have helped us to improve the quality of our manuscript.

We really do hope that the corrections included will provide a satisfactory answer to your comments.

Sincerely,

Ricardo Soto-Rifo

Reviewer 1

The manuscript by Hernandez-Diaz et al. is an interesting and timely review of DDX3/ human virus interplay. The authors describe DDX3 physiological functions in RNA metabolism and innate immune signaling. Then they review how these diverse functions are hijacked and manipulated by RNA and DNA viruses. They describe the interactions engaged between DDX3 and viral proteins and whether they facilitate or instead control virus multiplication. These aspects are clearly synthesized and include recent data from the literature. The review is also very well illustrated, with clear figures that capture the mains aspects described in the review. In this regard, the review is of interest to people interested in the role of this host helicase in viral infections. 

Thank you very much

I only have minor suggestions, some concerning the Figures, others about the text.

Main remarks 

The text is very synthetic and some information is lacking. To be fully exhaustive, the list of viruses interacting with DDX3 during their replication cycle should be completed with data dealing with the role of DDX3 in enteroviruses (FMDV), paramyxoviruses (HPIV-3), and alphaviruses (VEEV) life cycle. 

We have included FMDV, HPIV-3 and VEEV in the revised version as suggested by the Reviewer

For each viral family, whether the reported role of DDX3 relies on its helicase activity or not should be indicated when possible. 

We have included this information in the revised version as suggested by the Reviewer

In the last paragraph, dedicated to DDX3 targeting for therapeutic interventions, it is not clear how anti-DDX3 molecules discussed in this section are acting. They are referred to as “DDX3 inhibitors”. Whether these molecules inhibit DDX3 helicase activity or are inhibiting other DDX3-associated functions needs to be mentioned as well as they potential interest to fight viruses discussed in the previous section. 

We have clarified this point in the text as suggested by the Reviewer

Given DDX3's pivotal physiological role in cell RNA fate, an important point relates to the possibility to inhibit DDX3 for antiviral purposes without general adverse effect. The final discussion needs to consider this point. 

We have included a small paragraph on this point in the concluding remarks section of the revised version as suggested by the Reviewer

Minor remarks on the Figures:

Since the review is focused on DDX3, Figure 1 should depict DDX3 organization instead of that of DEAD-box helicases in general. This especially refers to the exact organization of the N- and the C-terminus domains of DDX3 that should be represented instead of the variable domains indicated in the current figure. The size of each domain should be indicated, as well as binding domains for interacting proteins. The corresponding text needs to be modified accordingly “The N- and the C-terminus domains are variable amongst DEAD-box helicases ...”. 

We have modified the figure as suggested by the Reviewer

For more clarity, a Table could be inserted to summarize DDX3 pro- and anti-viral activities in the various models discussed in the review. The nature of viral proteins or nucleic acids interacting with this helicase could be inserted. 

We have included a Table as suggested by the Reviewer

Reviewer 2 Report

The manuscript nicely summarizes the current knowledge on the functional implications of cellular DDX3 in viral infections. The authors review the strategies used by different viruses to exploit the different functionalities of DDX3 to their own advantage. It is a comprehensive and concise review.

It is well written, however a slight reorganization of the information and especially the figures could improve the quality of the manuscript.

  • The position of figure 2. Roles of DDX3 in mRNA metabolism could be moved to page 3, just before the first paragraph (Indeed, DDX3 may….). Its current position breaking the information of section 3. Role of DDX3 in immune signaling pathways, does not make much sense and is confusing.
  • Similarly, Figure 4 Participation of DDX3 during HCV replication, currently located in the middle of the Human immunodeficiency virus section, could be moved to page 6 right after the paragraph “More detailed evidence….”
  • Subsection 4.1 Arenavirus should be moved to page 8, near to the information corresponding to Vaccinia virus, Herpex virus and HBV, and refer to figure 6. In fact Arenavirus are included in figure 6 along with Vaccinia virus, and HBV etc.

The acronym PAMP is missing in the Acronyms list.

Finally, are the figures original or are they copied or adapted from other publications? If the latter, this should be indicated in the figure legends and the original articles should be properly cited.

Author Response

Dear Reviewer,

On behalf of the authors and myself, I would like to thank for your comments that have helped us to improve the quality of our manuscript.

We really do hope that the corrections included will provide a satisfactory answer to your comments.

Sincerely,

Ricardo Soto-Rifo

Reviewer 2

The manuscript nicely summarizes the current knowledge on the functional implications of cellular DDX3 in viral infections. The authors review the strategies used by different viruses to exploit the different functionalities of DDX3 to their own advantage. It is a comprehensive and concise review.

It is well written, however a slight reorganization of the information and especially the figures could improve the quality of the manuscript.

  • The position of figure 2. Roles of DDX3 in mRNA metabolism could be moved to page 3, just before the first paragraph (Indeed, DDX3 may….). Its current position breaking the information of section 3. Role of DDX3 in immune signaling pathways, does not make much sense and is confusing.
  • Similarly, Figure 4 Participation of DDX3 during HCV replication, currently located in the middle of the Human immunodeficiency virus section, could be moved to page 6 right after the paragraph “More detailed evidence….”
  • Subsection 4.1 Arenavirus should be moved to page 8, near to the information corresponding to Vaccinia virus, Herpex virus and HBV, and refer to figure 6. In fact Arenavirus are included in figure 6 along with Vaccinia virus, and HBV etc.

The acronym PAMP is missing in the Acronyms list.

Finally, are the figures original or are they copied or adapted from other publications? If the latter, this should be indicated in the figure legends and the original articles should be properly cited.

Thank you very much for these comments. We have reorganized the text as suggested by the Reviewer. We have also added the acronym PAMP and indicated the figure that were adapted from a previous work.